# Inkjet-printed optical interference filters

Qihao Jin ®[1] ✉, Qiaoshuang Zhang ®[1], Christian Rainer[1,2], Hang Hu ®[1,3], Junchi Chen[1], Tim Gehring ®[1], Jan Dycke[1], Roja Singh ®[1,3], Ulrich W. Paetzold ®[1,3], Gerardo Hernández-Sosa ®[1,2,3], Rainer Kling[1] & Uli Lemmer ®[1,2,3] ✉

Optical interference filters (OIFs) are vital components for a wide range of optical and photonic systems. They are pivotal in controlling spectral transmission and reflection upon demand. OIFs rely on optical interference of the incident wave at multilayers, which are fabricated with nanometer precision. Here, we demonstrate that these requirements can be fulfilled by inkjet printing. This versatile technology offers a high degree of freedom in manufacturing, as well as cost-affordable and rapid-prototyping features from the micron to the meter scale. In this work, via rational ink design and formulation, OIFs were fully inkjet printed in ambient conditions. Longpass, shortpass, bandpass, and dichroic OIFs were fabricated, and precise control of the spectral response in OIFs was realized. Subsequently, customized lateral patterning of OIFs by inkjet printing was achieved. Furthermore, upscaling of the printed OIFs to A4 size ($29.7 \times 21.0$ cm$^2$) was demonstrated.

Optical interference filters (OIFs) are indispensable for numerous optical and photonic systems. To highlight some, vast applications are found in, e.g., nonlinear optics[1], biochemical & biomedical optics[2], imaging systems[3], laser systems[4], and micro-cavity-based photonic devices[5]. OIFs consist of multiple thin film layers with thicknesses ranging from tens to even hundreds nanometers[6–8]. They utilize optical interference to achieve a desired transmission or reflection in the optical band of interest[9], which is challenging to fabricate and principally different from color filters (Supplementary Table 1). The quality of the optical response of OIFs relies on precise and homogeneous thin film deposition, requiring a thickness deviation of each layer below 3%[10,11]. These facts pose numerous technical challenges in OIF manufacturing. Nowadays, the fabrication of high-end OIFs is only conducted in high vacuum conditions and by technologies like electron beam evaporation, plasma-assisted sputtering, and atomic layer deposition[12]. All these methods rely on cost-intensive equipment and suffer from long deposition times. In addition, rapid prototyping of customized OIFs presents a severe obstacle for these methods. Moreover, some materials possess desirable properties for producing novel OIFs[13–15]; however, utilizing vacuum-based approaches to deposit them is unsuitable. In contrast, inkjet printing is an additive manufacturing method that allows ultraprecise thin film deposition with superior material efficiency[16]. Furthermore, it can be applied on large

areas and flexible substrates under ambient conditions. Despite these advantages, inkjet printing high-end OIFs such as bandpass filters have not been realized so far. The challenge lies in formulating inks and developing a printing process for achieving a highly repeatable deposition of homogenous layers. Especially when a large number of layers is required in order to realize a demanded optical property[17].

In response to this open challenge, this work demonstrates fully inkjet-printed inorganic OIFs, including longpass, shortpass, bandpass, and dichroic filters. The OIFs were digitally fabricated in ambient conditions using a desktop inkjet printer. It was achieved via rational ink design and the developed printing process. Lateral patterning of OIFs was also realized for emerging solar applications. Moreover, an industrially applicable route for upscaling inkjet-printed optical filters is demonstrated by printing dichroic filters in A4 ($29.7 \times 21.0$ cm$^2$) size.

## Results

### OIF fabrication, ink formulation, and printability characterization

Until now, high-end OIFs consisting of multilayer structures are fabricated by vacuum-based technologies. In this work, such OIFs with different sizes and optical properties were inkjet printed in ambient (Fig. 1a). The printed OIFs comprise alternately stacked thin films with high and low refractive index (RI) on each other (Fig. 1b). Inorganic-

[1]Light Technology Institute (LTI), Karlsruhe Institute of Technology (KIT), Engesserstrasse 13, 76131 Karlsruhe, Germany. [2]InnovationLab, Speyerer Strasse 4, 69115 Heidelberg, Germany. [3]Institute of Microstructure Technology (IMT), Karlsruhe Institute of Technology (KIT), Hermann-von-Helmholtz-Platz 1, 76344 Eggenstein-Leopoldshafen, Germany. ✉e-mail: qihao.jin@kit.edu; uli.lemmer@kit.edu

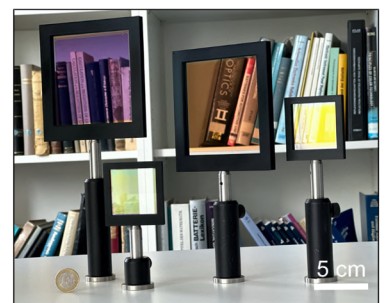 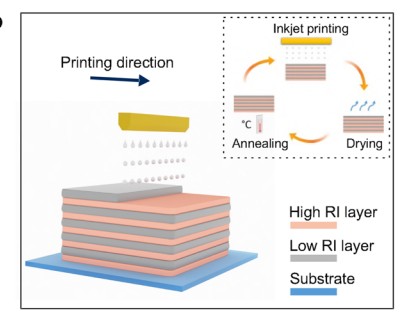 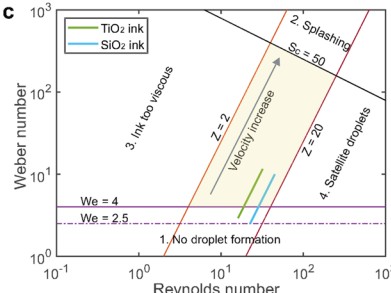

**Fig. 1 | OIFs fabrication and ink printability characterization. a** Inkjet-printed OIFs in holders. **b** Schematic illustration of multilayer inkjet printing and a complete printing cycle. **c** Ink printability analysis.

nanoparticle-based inks were used for fabricating layers with different refractive indices. Silicon dioxide ($SiO_2$) and titanium dioxide ($TiO_2$) are the prevalent materials in the optical filter manufacturing industry. We therefore used nanoparticulate $SiO_2$ (amorphous phase) and $TiO_2$ (rutile phase) in this work. The complete printing cycle contains the processes of inkjet printing, drying, and annealing (Fig. 1b inserted). To fabricate OIFs with designed and homogenous optical properties, inks were carefully formulated and assessed for the printing process. The detailed ink design and development are described in the Supplementary Information (Supplementary Figs. 1–8). It commences with surface free energy (SFE) characterization of the $SiO_2$ and $TiO_2$ solid surfaces. Subsequently, it continues with design and employment of proper organic vehicles for formulating nanoparticle-based inks. Thereafter, characterization of ink surface tension (SFT) and the printability was conducted. Ultimately, the RIs of printed solid layers were measured for designing multilayer structures to achieve the demanded optical properties in OIFs. All the steps mentioned above were interrelated during ink design and optimization.

In this work, the ink printability characterization is an important task. It allows us to analyze the performance of the developed inks. Throughout the inkjet printing, SFT, density, and dynamic viscosity of inks affect the physics and fluid mechanics in droplet ejection and formation[18]. Over the last decades, many researchers have investigated this subject[18,19]. In this work, two methods are applied in parallel to validate the printability of the inks. The first method is based on three dimensionless values, namely Reynolds (Re), Weber (We), and Ohnesorge number (Oh)[20]:

$$Re = \frac{v\rho d}{\eta} \qquad (1)$$

$$We = \frac{v^2\rho d}{\gamma} \qquad (2)$$

$$Oh = \frac{1}{Z} = \frac{\sqrt{We}}{Re} = \frac{\eta}{\sqrt{\gamma\rho d}} \qquad (3)$$

where $v$ is the ejected droplet velocity (in m s$^{-1}$), $d$ is the characteristic length (in m, which is the nozzle orifice size). $\gamma$, $\eta$, and $\rho$ are the surface tension (in N m$^{-1}$), dynamic viscosity (in Pa s), and density (in kg m$^{-3}$) of the ink, respectively. The ink printability is evaluated by four boundaries (Fig. 1c). The first boundary is defined by the We of 4, where the region We < 4 means there is no sufficient energy to form a droplet[21]. This is mainly because the fluid has insufficient energy to overcome its surface tension to be ejected from the nozzle. The second and third boundaries are defined by $Z = 2$ and $Z = 20$, respectively[22], where $Z$ is the inversion of Oh. The region $Z < 2$ denotes that the ink is too viscous, while $Z > 20$ implies the formation of satellite droplets instead of single droplets during ink jetting. The fourth boundary is defined by the

critical droplet velocity $S_c$. Above this velocity, splashing occurs when droplets hit the substrate surface[23].

$$S_c = We^{1/2}Re^{1/4} \qquad (4)$$

According to previous work[23], the onset of splashing happens when $S_c > 50$. Hence, a Re–We space diagram is built, and a printable region is confined by the four aforementioned boundaries. Inks with properties fitting in this printable region are considered suitable for inkjet printing. According to the measured ink viscosity (2.0 mPa s of $SiO_2$ ink and 2.5 mPa s of $TiO_2$ ink) and droplet velocities, the corresponding line of each ink is plotted (Fig. 1c). The length of each ink line is defined by the velocity range of ejected droplets without satellite lines (2–4 m s$^{-1}$). At lower droplet speed, the We of our $SiO_2$ ink reaches a minimum value of 2.5. The We of the $TiO_2$ ink also shows a value below 4. The lower We value and droplet velocity indicate the lower energy required to eject ink droplets from the nozzles. Depending on the location of both lines, inks are considered suitable for inkjet printing. Additionally, a second method (Supplementary Fig. 2) is also applied in this work to complement the evaluation of the ink printability[22]. The results imply the suitability of the inks for inkjet printing, which agrees with method one. For clarity, the rheological properties of developed inks are summarized (Supplementary Table 2). In addition to the ink formulation and printability investigation, the printing process was also optimized in this work for achieving homogenous layers. The impact of substrate temperature and printing speed are investigated and optimized (Supplementary Figs. 7 and 8).

### Inkjet-printed longpass, shortpass, and bandpass OIFs
Longpass and shortpass filters are also known as edge filters due to their ability to exhibit a sharp transition in transmission within a narrow spectral range. A common way to design edge filters starts with quarter-wave (QW) stacks. However, a significant challenge in design and fabrication is to diminish the ripples in the pass region, which result from the periodic-structure nature of the QW stacks. The strategy to ease these ripples is by optimizing individual layer thickness. As a result, parts of the complete layer structure become nonperiodic. In such nonperiodic structures, the thickness difference between each layer can vary from several to some hundred nanometers[24]. Therefore, it is a particular challenge for inkjet-printed OIFs to precisely control the required thickness and deviation. In this work, we report a well-controlled digital inkjet printing process, which allows the fabrication of a large number of layers with individually optimized thicknesses. This was achieved by printing each layer in one printing cycle with digital control of the printing resolution (Supplementary Fig. 9). The results show a good linear dependency of the printing resolution and the deposited layer thickness, which is quantitatively indicated by a coefficient of determination of 0.9999. Moreover, the reproducibility of the printing process was also investigated. Surface roughness of the printed layers was characterized by an atomic force microscope,

showing low root-mean-square roughness values (Supplementary Fig. 10). The results indicate that the printing of OIFs is consistent and reproducible, as evidenced by the analysis of their optical performances (Supplementary Fig. 11).

The concept of a longpass filter is presented in the schematic (Fig. 2a), where the longer wavelengths transmit through the filter. The scanning electron microscope (SEM) image (Fig. 2b) shows the cross-section of the inkjet-printed longpass filter, where the filter was printed on a glass substrate. The color stack on the right side shows a schematic of the device. The layer thickness difference in the nonperiodic structure is also depicted in the color stack, where the low and high refractive index materials are shown in different colors. The glass substrate is in black. Both the simulated (Fig. 2c) and measured (Fig. 2d) transmitting spectra show lessened ripple compared to a periodic structure (Supplementary Fig. 12a). In a longpass filter, the cut-on wavelength refers to the wavelength at which the transmittance of the filter reaches 50% of its maximum value. Since the maximum transmittance for the printed longpass filter is 94.4% at 714 nm, the cut-on wavelength is 532 nm, accordingly. Besides, the average transmittance between 540 and 800 nm is 91.7%. The diffuse share in total transmittance was also measured and plotted as the gray dashed line (Fig. 2d). The value remains below 2% between 600 and 800 nm. It increases slightly in the range of 500–600 nm, which can be assigned to the higher extinction of the $TiO_2$ layer at shorter wavelengths.

The concept of a shortpass filter is depicted (Fig. 2e), where the shorter wavelengths pass through the filter. The SEM image and the color stack reveal the nonperiodic structure (Fig. 2f). Simulated (Fig. 2g) and measured (Fig. 2h) transmittances indicate a pass region between 400 and 600 nm. Moreover, the ripples in the pass region are also lessened compared to the periodic structure (Supplementary Fig. 12b). In a typical shortpass filter, the cut-off wavelength is the wavelength where transmittance reduces to 50% of the maximum. The ripples are slightly more visible in our shortpass filter than in the longpass filter over the pass region. Given this context, the cut-off wavelength in the measured spectrum, 544 nm, is therefore calculated based on the average transmittance between 400 and 533 nm, which amounts to 77.4%. In addition, the diffuse transmission in the shortpass filter reduces from 5% to nearly zero from 400 to 600 nm.

Based on the successful realization of the long- and shortpass filters, a bandpass filter was printed by combining the two units. The concept of the bandpass filter is shown (Fig. 2i). It only allows the selected bandwidth to pass through. The structure of the printed bandpass filter is presented on the right side of the same figure, showing that the shortpass filter was first printed on a glass substrate and followed by printing the longpass filter on top. The SEM image shows the cross-section of the printed bandpass filter (Fig. 2j), which consists of 39 single layers in total. The longpass part, shortpass part, and intermediate buffer layer are indicated with red, blue, and green colors, respectively. Here, the buffer layer consists of two layers of the same material. From the simulated (Fig. 2k) and measured (Fig. 2l) transmission spectra, the peak wavelength of the bandpass filter is found to be 537 and 535 nm, respectively, showing a good realization from the design to the fabrication. The maximum measured transmittance of 53.5%, however, is lower than the 72.3% found for the

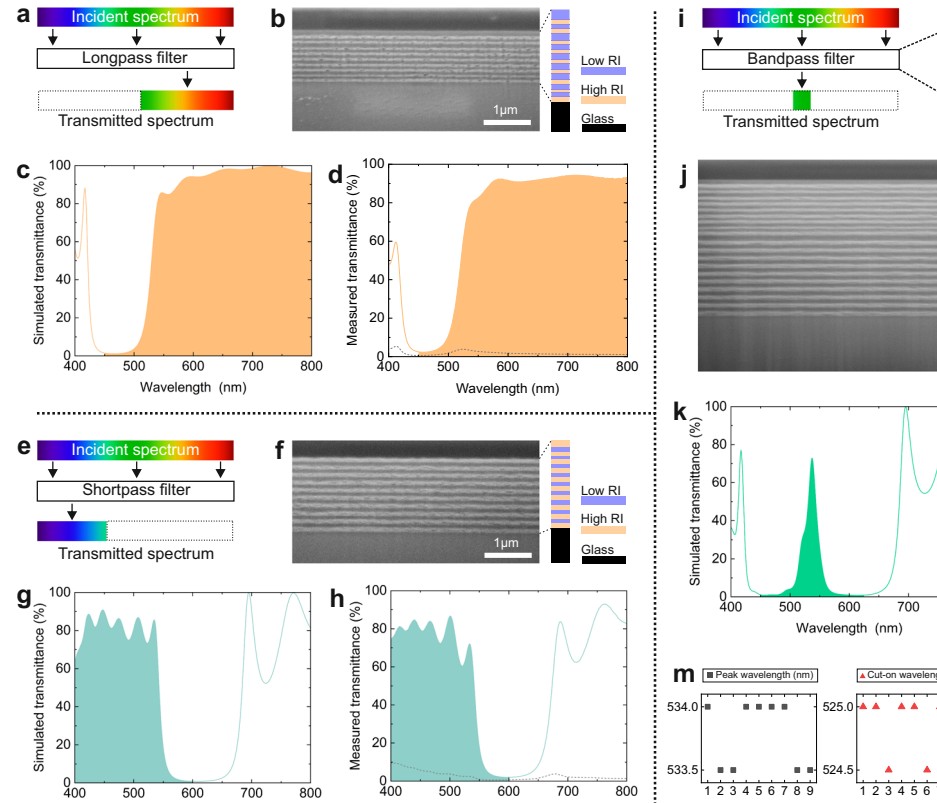

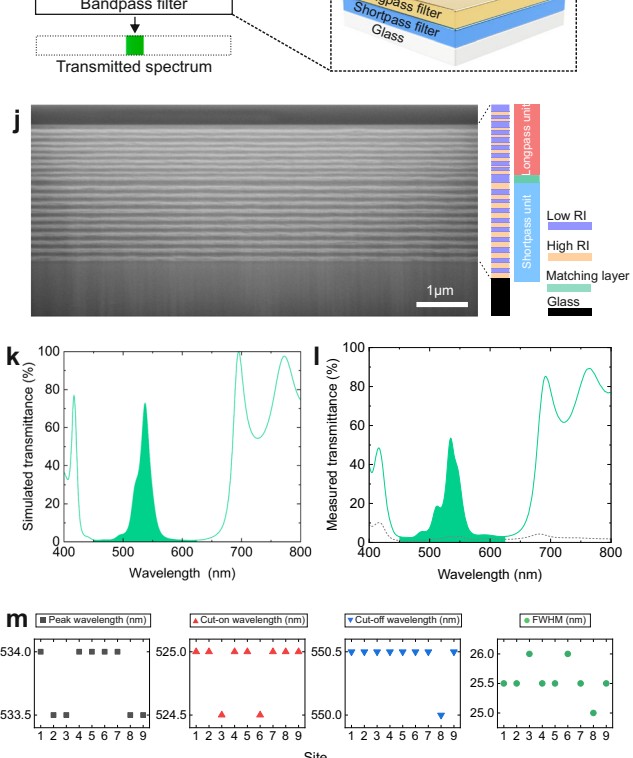

**Fig. 2 | Printed all dielectric longpass, shortpass, and bandpass OIFs. a** Concept of a longpass filter. **b** SEM cross-section view of the printed filter. **c** and **d** Simulated and measured transmittance of the printed longpass filter, respectively. The dashed gray line is the diffuse transmittance. **e** Concept of a shortpass filter. **f** SEM cross-section view of the printed filter. **g** and **h** Simulated and measured transmittance of the printed shortpass filter, respectively. The dashed gray line is the diffuse transmittance. **i** Concept and design of a bandpass filter. **j** SEM cross-section view of the printed filter. **k** and **l** Simulated and measured transmittance of the printed bandpass filter, respectively. The dashed gray line is the diffuse transmittance. **m** Characterization of optical homogeneity in the printed bandpass filter, including peak, cut-on, cut-off wavelength, and full width at half maximum (FWHM). The color stacks next to the SEM images provide an illustration of the layer scheme and functionality. Different colors indicate materials with low or high refractive index (RI), as well as the glass substrate.

simulated stack. This difference very likely results from the slight blue shift of the shortpass spectrum in the fabricated stack, which can be further deduced from the blue-shifted peak wavelength of the bandpass filter. According to the maximum transmittance and the center wavelength, the cut-on and cut-off wavelength in the printed filter are 526 and 551 nm, respectively. Furthermore, the diffuse transmittance was measured, and it remains below 3% in the region of interest, which is from 450 to 630 nm.

Moreover, the peak, cut-on, and cut-off wavelengths were measured on nine sites of the sample to characterize the optical homogeneity in the printed bandpass filter for a half-inch-size surface region. The nine sites were arranged as a 3-by-3 array (Supplementary Fig. 13). The spot diameter during the measurement was 2 mm with a center-to-center distance of 3 mm. Full width at half maximum (FWHM) values were also extracted from the measured spectra, giving an average value of 25.5 nm. The individual values of peak, cut-on, cut-off wavelength, and FWHM are shown in the respective diagram (Fig. 2m). The variations of the first three parameters are only 0.5 nm, and the variation in FWHM is 1 nm. These minor variations evidence the high uniformity of our printed bandpass filter.

As an alternative, bandpass filters can also be fabricated based on the concept of a Fabry–Pérot (FP) etalon, offering a common bandpass property (Fig. 3a). It can be designed as a resonator consisting of two mirrors (M1 and M2) and one cavity (Fig. 3b). When the cavity has an appropriate thickness, constructive interference of light beams will result in a transmission spectrum with a narrow bandwidth. In this work, QW stacks were printed and used as highly reflecting mirrors due to their low optical loss. Each printed QW stack herein comprised seven single layers, and the cavity material was $SiO_2$. The SEM cross-section (Fig. 3c) and the color stack present the detailed layer structure of the printed component. The mirrors are depicted with gray color while the cavity is in green. The peak wavelength in the simulated (Fig. 3d) transmission spectrum matches well with the measured (Fig. 3e) one in the band of interest, which is 580 and 577 nm, respectively. Furthermore, the transmittance at the peak wavelength is 85.9% and 83.7% in simulation and fabricated filters, respectively, implying a good realization of design-to-device. Additionally, the cut-on and cut-off wavelengths and the FWHM are measured as 566, 590, and 24 nm, respectively. The diffuse transmittance is below 5% in the bandpass region (Fig. 3e, dashed line). As a common strategy to suppress the unwanted sideband transmittance, a silver layer was included in the cavity (Fig. 3f). The silver layer was thermally evaporated with a thickness of 60 nm. After adding this silver layer, the sideband transmittance is dramatically suppressed (Fig. 3g and h). The cut-on, cut-off

wavelength and the FWHM are 566, 577, and 21 nm, respectively. The peak transmittance was measured to be 40%. This reduced peak transmittance is assigned to the absorption inside the silver layer[12] and the surface plasmons at the metal-dielectric interfaces[25]. However, utilizing the FP etalons allows for a decrease in the number of layers for realizing a narrow bandwidth transmittance compared to the filters based on all-dielectric materials.

## Tunability, patterning, and upscaling of inkjet-printed OIFs

To show the good control of optical response in OIFs using inkjet printing, various dichroic filters (consisting of 9 bi-layers) with different optical characteristics were printed. Filters were designed with different reflecting center wavelengths, i.e., 380, 480, 590, 620, and 680 nm, respectively (Fig. 4a). The measured values for the printed filters were 383, 479, 592, 619, and 678 nm, respectively (Fig. 4b). The minor variations in center wavelength indicate not only a precise control of the printing process but also a highly customizable optical response in inkjet-printed filters. In the following, we demonstrate the patterning of OIFs. A pattern was printed, giving a vivid color that is generated from optical interference (Fig. 4c). As an emerging use case, a building-integrated photovoltaic model based on a colored solar panel is also demonstrated (Fig. 4d), where the color is the reflected spectrum of the patterned OIFs. The house was 3D-printed, and two silicon solar panels were installed on the same side of the roof. The left side of the roof shows the solar panel without OIFs, while the right side shows another panel covered with laterally patterned OIFs.

As another essential step towards industrial manufacturing, upscaling of the printed OIFs is further demonstrated in this work. $10 \times 10 cm^2$ dichroic filters were printed as a first step to validate the suitability of our inks and the printing process. In the optical images, the filter reflects the green part of the spectrum (Fig. 4e, dark background) while the red light is transmitted (Fig. 4f, bright background). To further show the spectral splitting functionality, the filter was placed on a table facing the sun (Fig. 4g). A one-euro coin and a caliper were placed aside as scale references. $10 \times 10 cm^2$ size filters with different spectral responses were additionally printed and mounted in holders (Fig. 4h). The reflection spectrum of the filter in Fig. 4e was measured and normalized to the peak wavelength of 515 nm (Supplementary Fig. 14). Twenty individual sites on the filter (Supplementary Fig. 15) were measured for rise-on and fall-off wavelengths, which are the wavelengths correspond to 50% of the peak reflectance, as well as FWHM to characterize the optical uniformity of the upscaled dichroic filter (Supplementary Fig. 16). The results show that the variations are only 3 nm (min. 448 nm, max. 451 nm) for the rise-on wavelengths,

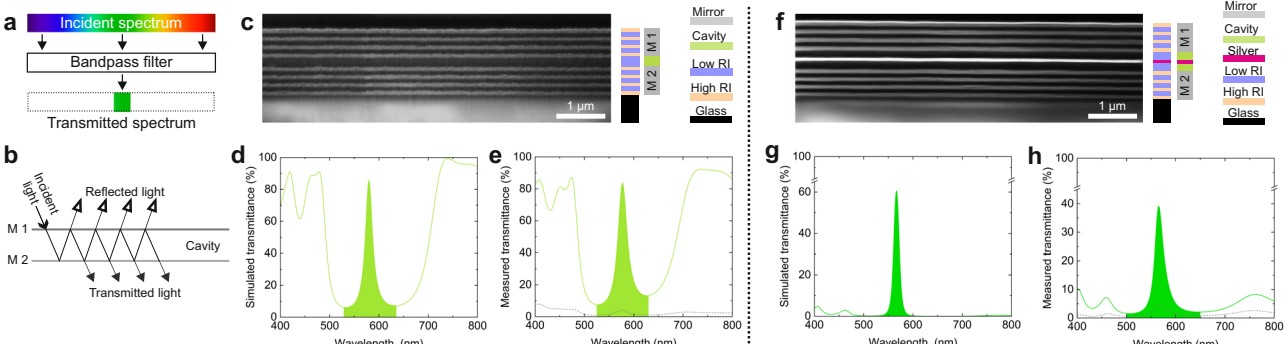

**Fig. 3 | Printed bandpass OIF based on Fabry–Pérot etalons. a** and **b** Concept and principle of a Fabry–Pérot etalons type bandpass filter. The structure consists of mirror 1 (M1), mirror 2 (M2), and a cavity. **c** SEM cross-section view of the printed filter. **d** and **e** Simulated and measured transmittance of the printed filter, respectively. The dashed gray line is the diffuse transmittance. **f** SEM cross-section view of the printed filter after integrating a silver layer. **g** and **h** Simulated and measured

transmittance of the printed filter, respectively. The dashed gray line is the diffuse transmittance. The color stacks next to the SEM images provide an illustration of the layer scheme and functionality. Different colors indicate materials with low or high refractive index (RI), as well as the glass substrate. The green block indicates the cavity, and red denotes the silver layer.

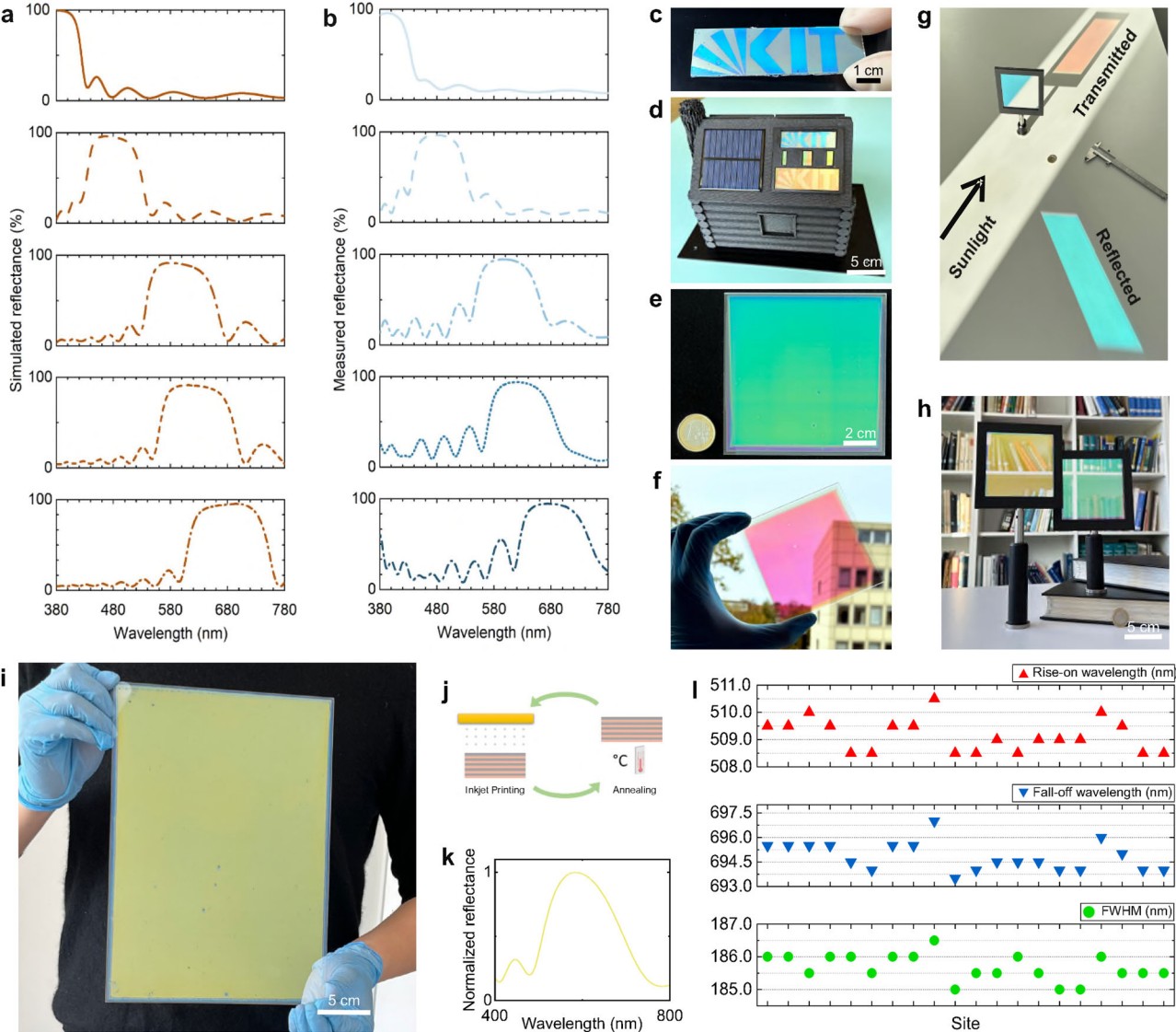

**Fig. 4 | Spectrum tuning, upscaling, and patterning of OIFs. a** and **b** Simulated and measured reflectance tuning of OIFs in visible wavelength range, respectively. **c** Lateral patterned OIF. **d** OIFs on the solar panel for a solar house. **e** and **f** Printed 10 × 10 cm² dielectric filter in the dark and bright background, respectively. One Euro coin is a scale reference. **g** Spectrum splitting capability in OIF under sunlight illumination. **h** Printed 10 × 10 cm² OIFs installed in holders. **i** Printed OIF in an A4 size. **j** Printing cycle developed for large-size OIF printing. **k** and **l** Normalized reflectance and uniformity characterization of optical performance in printed A4 OIF, respectively. Results show the rise-on, fall-off wavelength, and full width at half maximum (FWHM).

3 nm (min. 606.5 nm, max. 609.5 nm) for the fall-off wavelengths, and 1 nm (min. 158 nm, max. 159 nm) for FWHM, respectively. These slight variations indicate a homogenous spectral response in the printed 10 × 10 cm² dichroic filters. To show the potential of our inks and the printing process for further up-scaling, a dichroic filter was printed in DIN A4 (21.0 × 29.7 cm²) size (Fig. 4i). This is also the maximum size of printed areas possible for our inkjet printer. A highlight in the large-size printing, both in 10 × 10 cm² and A4, is that no individual drying process is needed in the printing cycle (Fig. 4j). This was achieved by carefully adjusting the printing speed so that the printed wet layers were able to be dried simultaneously during printing. The reflection spectrum of this A4-size filter was also measured and normalized to the peak wavelength of 580 nm (Fig. 4k). Rise-on and fall-off wavelengths and FWHM were also characterized (Fig. 4l) over twenty individual sites on the filter (Supplementary Fig. 17). The results show a good uniformity of the optical performance, where the variations of the rise-on, fall-off wavelengths, and FWHM are 2 (min. 508.5 nm, max. 510.5 nm), 2.5 (min. 693.5 nm, max. 696 nm), 1.5 (min. 185 nm, max.

186.5 nm) nm, respectively. These results indicate that upscaling of the filters can be done without compromising the optical quality. We mention that this also opens the possibility to cut customized smaller size filters from the larger area, e.g., by laser cutting.

## Discussion

In summary, based on rational ink design and optimized printing processes, optical interference filters (OIFs) were fully inkjet printed in ambient conditions. Longpass, shortpass, and bandpass filters were fabricated. The individual spectral responses matched well with the initial simulation, showing a promising realization of a customized design. The developed inks and processes were applied to realize the spectral tunability, lateral patterning, and upscaling of the printed OIFs. A dielectric filter with DIN A4 size was inkjet printed. Additionally, we exhibited the versatility of this developed method by fabricating a flexible and patterned filter on a foil, achieved by straightforward adjustments of the process parameters (Supplementary Fig. 18). Subsequently, the strategy to further improve the quality of the inkjet-

printed filters is given. For instance, the suppression of residual transmittance in edge and bandpass filters via the integration of additional optical cavities (Supplementary Fig. 19) and enhancing the optical density by increasing the total number of layers in OIFs (Supplementary Figs. 20–22). Moreover, our printed OIFs show good durability based on the results of adhesion and abrasion tests (Supplementary Tables 3 and 4, and Supplementary Fig. 23). We expect this work to be an important step towards an industry-relevant method of producing OIFs with lower cost in manufacturing, variable size to meet different demands, and higher flexibility in designing and implementation. We foresee various applications in the fields of advanced optical instrumentations[26], solar energy[27], and clinical applications[28].

## Methods

### Surface characterization

The contact angle (CA) was measured by a CA measuring system (OCA 50, DataPhysics Instruments) in a static state under ambient conditions. The analysis was done using the Sessile-Drop method and the Laplace–Young equation. For calculating the surface free energy (SFE) of a particular solid surface, deionized $H_2O$, ethylene glycol (Sigma-Aldrich), and diiodomethane (Sigma-Aldrich) were used. The OWRK method[29] was applied to calculate the polar and dispersive part of the SFE. The $SiO_2$ ($TiO_2$) solid surface in the CA measurements was fabricated by spin-coating $SiO_2$ ($TiO_2$) nanoparticle dispersion to form a thin film with a thickness of around 100 nm. It was followed by subsequent annealing at 250 °C for 5 min. The thickness of the thin film was measured by a profilometer (DektakXT, Bruker). On each solid surface, 10 different sites were used to measure the CA of the liquid (single dosing volume of 0.5 μl), and the average CA was used in the SFE calculation. The average CA of $SiO_2$ ($TiO_2$) ink on $TiO_2$ ($SiO_2$) surfaces was obtained by measuring 10 different sites as well.

### Characterization of ink properties

The total liquid/ink surface tension (SFT) was measured by an SFT measuring system (OCA 50, DataPhysics Instruments) under ambient conditions. The analysis was done in the Pendant Drop method, using the Laplace–Young equation. To obtain the polar and dispersive share in the total SFT of a liquid, polyethylene (Thermo Scientific) was used as a purely dispersive reference surface. The viscosity of the ink was measured using a viscometer (m-VROC, RheoSense).

### Nanoparticle ink formulation

MEK-$SiO_2$ nanoparticle dispersion (Nissan Chemical) was added to a glass flask. Due to the high concentration and low boiling point of MEK, the dispersion dried promptly, leaving only the solid $SiO_2$ nanoparticles. 1,3-dimethoxybenzene was then added into the flask to form $SiO_2$ ink (concentration of 3 wt%). The $TiO_2$ nanoparticle dispersion (Avantama) was mixed with 2-propoxyethanol (Sigma-Aldrich) with a weight ratio of 1:9 to form $TiO_2$ ink (concentration of 1.8 wt%). The $SiO_2$ nanoparticles are in the amorphous phase, and the $TiO_2$ nanoparticles are in the rutile phase (Supplementary Fig. 24).

### XRD, SEM, optical, and AFM characterization

The phase components of the nanoparticles were characterized by X-ray diffraction (XRD). It was performed by a Bruker D2Phaser system with Cu-Kα radiation ($\lambda = 1.5405$ Å) in Bragg–Brentano configuration using a LynxEye detector. Refractive indices and extinction coefficients of the materials were measured by ellipsometry (VASE ellipsometer, J.A. Woollam). Scanning electron microscope cross-section images were obtained by SUPRA 55 (Carl Zeiss) at 2 kV. Focused ion beam (Zeiss Crossbeam 1540 EsB) was used for milling the sample to enable SEM image acquirement. Transmittance and reflectance were measured with a spectrophotometer (Lambda 1050 UV–vis–NIR, PerkinElmer) equipped with an integrating sphere. The transmittance was obtained with the normal incident angle, and the reflectance was obtained with an 8° incident angle. The surface morphology was measured by atomic force microscope (AFM) (NanoWizard, Bruker Nano), and the surface RMS roughness was analyzed with the software Gwyddion.

### Design and simulation

The optical design and simulation of the printed filters were carried out using the software Essential Macleod. The optimization of the nonperiodic structure was performed using the integrated method Optimac. Both synthesis and refinement steps were involved in producing the final design.

### Substrate preparation

The $2.5 \times 2.5$ and $10 \times 10$ cm² glass substrates (Präzisions Glas & Optik GmbH) were cleaned in an ultrasonic bath in deionized water, acetone, and isopropanol for 10 min each. The A4 size glass substrate (Präzisions Glas & Optik GmbH) was manually cleaned with deionized water, acetone, and isopropanol. All the substrates were blown with nitrogen gas flow for drying and removing particles.

### Inkjet printing of OIFs

The optical interference filters were printed by an inkjet printer (PixDro LP50) equipped with 10 pL cartridges (Fujifilm Dimatix). The printing process was completed in an ambient atmosphere (temperature at 21–22 °C and humidity at 40–50%). Both inks were filtered by PTFE filters with a pore size of 0.2 μm before filling them into the ink tank. The printhead temperature was set to 27 °C for printing both inks. The waveforms were set for 2–2.5 kHz jetting frequency. The substrate's moving direction is on the $y$-axis during printing. Depending on the required layer thickness, the printing resolutions were in the range of 500–900 dots per inch (dpi). Longpass, shortpass, and bandpass filters were printed on $2.5 \times 2.5$ cm² glass substrates. The printing speed was 70–100 mm s$^{-1}$, depending on dpi. The quality factor was 2, and the jetting frequency was 2 kHz. The printer substrate temperature was set to 25 °C. $TiO_2$ layers were dried naturally in the ambient atmosphere. The $SiO_2$ layers were dried at a slightly reduced pressure, 0.8 atm, to increase the drying speed. Filters printed on a $10 \times 10$ cm² glass substrate were designed and printed in QW structures. The printing speed was 100–115 mm s$^{-1}$, depending on dpi. The quality factor was 2, and the jetting frequency was 2 kHz. The printer substrate temperature was set to 25 °C. Both $TiO_2$ and $SiO_2$ layers were dried naturally in the ambient atmosphere during printing. Filters printed on the A4 glass substrate were designed and printed in QW structures. The printing speed was 125 mm s$^{-1}$, the quality factor was 2, and the jetting frequency was 2.5 kHz. The printer substrate temperature was set to 25 °C. Both $TiO_2$ and $SiO_2$ layers were dried naturally in the ambient atmosphere during printing. All the annealing processes were performed using a hotplate with the temperature at 200 °C for 10 min.

## Data availability

The simulation data of the layer thickness generated in this study are provided in the Supplementary Information. All data are available from the corresponding author upon request.

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

## Acknowledgements

The work is supported by the Deutsche Forschungsgemeinschaft (DFG, German Research Foundation) under Germany's Strategy via the Excellence Cluster 3D Matter Made to Order (3DMM2O, EXC-2082/1-390761711). We also acknowledge support from the Karlsruhe School of Optics & Photonics (KSOP). The authors thank the Helmholtz Association MTET program (Materials and Technologies for the Energy Transition)—Topic 1—Photovoltaics (38.01.05). Part of this work has been supported by the Helmholtz Association in the framework of the innovation platform "Solar TAP". H. Hu and J. Chen would like to thank the financial support from the Chinese Scholarship Council (CSC). We acknowledge support by the KIT-Publication Fund of the Karlsruhe Institute of Technology.

## Author contributions

U.L. conceived the study. Q.J. performed ink formulation, CA, SFE, SFT, optical transmission, and SEM characterization. Q.Z. performed the design and simulation. Q.J. and Q.Z. performed the inkjet printing process. C.R. performed ink viscosity characterization. H.H. performed silver layer deposition. J.C. performed substrate preparation. R.S. performed the XRD characterization. Q.J. wrote the manuscript, T.G., J.D., U.W.P., G.H.-S., R.K. and U.L. contributed to manuscript preparation and editing.

## Funding

## Competing interests

The authors declare no competing interests.
