## [Peer Review File · Nature Communications]

Inkjet-printed Optical Interference FiltersREVIEWER COMMENTS

Reviewer #1 (Remarks to the Author):

The paper presents development and a thorough study of Inkjet printed optical interference filters. The paper may be of a substantial interest in the optics field.

The study contains thorough characterization of the SiO₂- and TiO₂- based inks via standard methods routinely used by e.g. InkJet manufacturers. The inks are suspensions of SiO₂ and TiO₂ nanoparticles in solvents routinely used in such InkJet inks. Both materials have broad spectrum of applications and there are numerous papers reporting TiO₂ and SiO₂ printed via InkJet, including working recipes for such inks (one example is perovskite solar cell field). Some examples: *Molecules* 2017, 22(11), 2020; *Macromolecular Research* volume 26, pages1123–1128 (2018); DOI: 10.1109/ICSENS.2012.6411297; *International Journal of Precision Engineering and Manufacturing-Green Technology* volume 8, pages445–451 (2021); <https://doi.org/10.1016/j.mssp.2018.03.015>; *Royal Society Open Science* 7 (4), 200242, 2020; etc., etc., etc.

The concept of printable optical filters has also been attempted before, at least for color filters.

In methods insufficient information is given about the nanoparticles. For example, what crystallographic type of TiO₂ (SiO₂) was used?

Cross-section images in Fig.2 are very low resolution, but even with that on e.g. 2(b) one can see quite a lot of defects and inhomogeneities, which raises a question of reproducibility that should be addressed. Insufficient information is given about the produced multilayers. E.g. numbers for thicknesses of the presented layers in the films should be given.

In line 196 "solar penal" should probably be corrected.

In supplementary information, in fig.10, the printed layer shows substantial inhomogeneities. There's a significant edge defect, as well as what looks like a thickness gradient. However, authors characterize homogeneity of only a fraction of the film, which appears more homogeneous. This doesn't appear statistically relevant.

Reviewer #2 (Remarks to the Author):

In the manuscript titled "Inkjet-Printed Optical Interference Filters", the authors employed inkjet printing to fabricate optical interference filters, including longpass, shortpass, and bandpass filters. Compared to electron beam evaporation, plasma-assisted sputtering, and atomic layer deposition, this approach is versatile, fast, and cost-affordable. Moreover, the thin film on a large-area substrate, e.g., A4 glass substrate, showed good optical homogeneity. I recommend acceptance of the manuscript if the following issues could be addressed.

1 What about the mechanical performance of the filter? Regarding various applications, the optical film must fulfill certain minimum mechanical requirements; these include good adhesion (evaluated using an adhesive tape peel test, frequently applied after sample exposure to a humid environment at elevated temperature), acceptable scratch and abrasion resistance to allow handling (for example, by performing a cheese cloth rubbing test).

2 The authors should compare the transmittance of the blocking range or wavelength pass range of their products with that of commercial OIFs or OIFs fabricated via vacuum-based approaches (fig. 3h). As far as I know, the optical performances of printed OIFs are still not good enough.

3 Detailed experimental results for the thickness control of the SiO₂ or TiO₂ layers should be

provided.

4 How did the distance between the glass substrates and the printer heads affect the filter quality?

5 The transmitting spectra of bare glass substrate should also be provided in the manuscript.

6 The authors mentioned that inkjet printing can be applied on large area and flexible substrates under ambient conditions. Since the annealing processes were performed using a hotplate with the temperature at 200 °C for 10 min, I wonder if the proposed method can be employed to fabricate flexible filters.

7 The authors should clarify if the optical multilayer thin films can be improved to eliminate the miscellaneous transmittance peaks of the OIFs, for example, longpass filter (400-450 nm) , shortpass filter (600-800 nm).

Reviewer #1:

The paper presents development and a thorough study of Inkjet printed optical interference filters. The paper may be of a substantial interest in the optics field.

Comment 1:

The study contains thorough characterization of the SiO₂- and TiO₂- based inks via standard methods routinely used by e.g. InkJet manufacturers. The inks are suspensions of SiO₂ and TiO₂ nanoparticles in solvents routinely used in such InkJet inks. Both materials have broad spectrum of applications and there are numerous papers reporting TiO₂ and SiO₂ printed via InkJet, including working recipes for such inks (one example is perovskite solar cell field). Some examples: *Molecules* 2017, 22(11), 2020; *Macromolecular Research* volume 26, pages1123–1128 (2018); DOI: 10.1109/ICSENS.2012.6411297; *International Journal of Precision Engineering and Manufacturing-Green Technology* volume 8, pages445–451 (2021); <https://doi.org/10.1016/j.mssp.2018.03.015>; *Royal Society Open Science* 7 (4), 200242, 2020; etc., etc., etc.

Reply to comment 1:

We thank the reviewer for the comment. SiO₂- and TiO₂-based inks have been reported for different applications. To the best of our knowledge, however, there is no previous report on using SiO₂- and TiO₂-inks for inkjet printing of high-precision optical interference filters. The existing reports address the use of TiO₂/SiO₂ for various applications, e.g., dielectrics and sensors. This is important work, but the requirements are much less demanding than what is needed for realizing optical interference filters. It is a huge effort to develop a proper inkjet printing process to use the formulated inks for **multilayer** interference filter printing. The challenges are the appropriate matching of many parameters, such as surface energy, surface tension, vapor pressure, viscosity, drying mechanisms, etc.

Regarding the perovskite solar cell field, to the best of our knowledge, there is no report on the inkjet-printed multilayer stacks for solar cell application yet. The group of C. Brabec has demonstrated blade-coated dielectric mirrors for the perovskite solar cell application, such as the work we referenced (doi: 10.1021/acsnano.6b00225). This is nice work but exhibits many qualitative and quantitative differences compared to our work. Blade coating is unsuitable for lateral patterning compared to inkjet

printing. Furthermore, the filter size demonstrated in this work ($210 \times 297 \text{ mm}^2$) is much larger than in the Reference work ($25 \times 75 \text{ mm}^2$). The reference work also did not show thickness control and ripple optimization of the spectrum. In Table 1 and Table 2, we compare the SEM image of the cross-section, size, and homogeneity of the printed filter fabricated in this work with the results presented in the Reference work.

Table 1. Comparison of SEM cross-section of the printed filters in this work and the reference work.

	SEM cross-section
This work (inkjet-printed, up to 39 layers)	Reference work (Blade-coated, up to 10 layers)	
Table 2. Comparison of the maximum size and homogeneity of the printed filters in this work and the reference work.

	Maximum size	Homogeneity
This work	$210 \times 297 \text{ mm}^2$	Reference work	$25 \times 75 \text{ mm}^2$	
The reviewer also listed papers that use SiO_2 - and TiO_2 -based inks for different applications that are not in the optics field. As optical filters rely on interference, precision on the scale of a small fraction of a wavelength is required. This is orders of magnitude more demanding than, e.g., a sensor layer.

For a better comparison, we include the papers mentioned by the reviewer in Table 3 and number them in the order of No. 1 to No. 6. We also include this work as **No. 0** in the same table. These numbers are used in Table 4 and Table 5 to indicate the corresponding work for convenience.

Table 3. List of different papers.

No.	Publication/work
0	This work
1	Molecules 2017, 22(11)
2	Macromolecular Research volume 26, pages1123–1128 (2018)
3	DOI:10.1109/ICSENS.2012.6411297
4	International Journal of Precision Engineering and Manufacturing-Green Technology volume 8, pages445–451 (2021)
5	https://doi.org/10.1016/j.mssp.2018.03.015
6	Royal Society Open Science 7 (4), 200242, 2020

In Table 4, we list the detailed information and parameters of each work. The comparison includes materials used, printed layer number, single layer thickness, whether the layer thickness resolution is controlled on a nanometer scale, the application that has been demonstrated, and the suitability for fabricating the optical interference filters (OIFs).

Table 4. Detailed comparison in different work.

No.	Material	Printed Layer number	Single Layer thickness	Layer thickness control on nm scale resolution	Demonstrated application	Suitability for OIFs considering layer quality
0	SiO₂ nanoparticles	Multilayer, Up to 39 layers	~50-80 nm	Yes	Optical interference filter	High
	TiO₂ nanoparticles		~50-120 nm			
1	Mesoporous silica nanoparticles	Single layer	~3.4 μm	No	Drug screening platform	Inappropriate
2	SiO ₂ hollow spheres	Single layer	~4.2 μm	No	Interlayer dielectrics	Inappropriate
3	SiO ₂ nanoparticles	Single layers	~100 nm (calculated from a layer by 5 times printing)	No	Humidity sensor	Inappropriate
4	Silica aerogel	Single layer	~3.5 μm	No	Thermal insulation layer	Inappropriate
5	TiO ₂ nanoparticles	Single layer	~140 nm	No	Material synthesis and characterization	low
6	SiO ₂ particle suspensions	Single layer	~70 nm	No	Material synthesis and characterization	Inappropriate
	TiO ₂ particle suspensions		~25 nm			

Compared to our work (**No. 0**), the work in publications No. 1 to No. 6 have not demonstrated the printability of multilayer stacks using the developed inks. Furthermore, all the work conducted in publications No. 1 to No. 6 does not demonstrate thickness control on a nm scale. In fact, the surface

morphology of the printed layers in publications No. 1 to No. 6 show very high roughness, thus being unsuitable for optical interference filters. We include the surface morphological information in Table 5 for a better illustration.

Table 5. Surface morphology comparison.

No. 0	SEM image: Top view of 39 layers 		AFM image: Top scan of 39 layers RMS roughness: 5.8 nm 		
No. 1	SEM image: Top view of 1 layer 	No. 2	SEM image: Top view of 1 layer (b) 		
No. 3	optical microscope image: Top view of 1 layer Printed layers in the red frame. Dimension of each printed dot: around 80 μm 		No. 4	SEM image: Top view of 1 layer (b) SiO₂ aerogel layer (After sintering) (c) 	
No. 5	SEM image: Top view of 1 layer a) 				
No. 6	Optical microscope image 	Top view of 1 layer Optical microscope image 	SEM image (a) 		

To clarify the novelty, our work shows the first-time inkjet-printed fully inorganic optical interference filters, which have not been demonstrated before.

The novelty includes the proof of concepts of:

1. Inkjet-printed dielectric interference filters
2. Homogenous multilayer large-area inkjet printing (up to A4 size: $210 \times 297 \text{ mm}^2$)
3. Up to 39-layer reproducible multilayer printing
4. Highly controlled layer thickness and its deviation (on a nanometer scale)
5. Highly controllable optical property
6. Patternable optical filters by inkjet printing
7. Printing task performed in ambient conditions

To emphasize the novelty, we included additional information in this work. The newly added illustration presents how thickness control is properly achieved.

- In the manuscript, new text is added:

The results show a good linear dependency of the printing resolution and the deposited layer thickness, which is quantitatively indicated by a coefficient of determination of 0.9999.

- In Supplementary information, a new section is added:

Thickness control of inkjet-printed layer

The strategy of thickness control in this work is realized by controlling the total amount of ink volume deposited on a unit area. The total deposited ink volume is governed by the number of printed droplets, controlled by the printing resolution, i.e., dots per inch (dpi).

Assume the number of printed droplets on a unit area is N_0 , resulting in a thickness of T_0 . For printing a thicker layer with a thickness of T_1 , a droplet number of N_1 is needed.

$$\begin{aligned} T_1 - T_0 &= \Delta T \\ N_1 - N_0 &= \Delta N \end{aligned}$$

A well-coupled relation between ΔT and ΔN is essential to realize good thickness control.

In this work, using formulated SiO_2 ink, different ink droplet numbers N_i (400, 500, 600, 700, 800, 900 thousand) were used to print on individual substrates with the same printing area of $25 \times 25 \text{ mm}^2$. Layer thickness was measured by a profilometer (Dektak, Bruker). For each N_i , printing was performed three times to show the repeatability of the thickness, which gives a standard deviation of the thickness. The results are shown in Supplementary Fig. 9.

It can be found that, with a ΔN of 100 thousand, the thickness increases linearly with the N . The coefficient of determination is 0.9999, indicating an excellent regression. Therefore, a well-controlled thickness by changing the ΔN can be successfully achieved. Furthermore, for each printed thickness,

individual deviation stays within 1% of the layer thickness, implying a highly reproducible thickness and a high repeatability of the printing process and results.

Supplementary Fig. 9. Thickness over the total amount of the printed droplet number N_i .

Comment 2:

The concept of printable optical filters has also been attempted before, at least for color filters.

Reply to comment 2:

We thank the viewer for the comment. In order to avoid misunderstandings here, we would like to point out that optical interference filters and color filters are based on completely different physical principles. The former is based on multilayer interference; the latter is based on absorptive materials such as dyes or color pigments. Color filters are widely used in color camera and display applications but do not rely on interference. They rely on the absorption in dyes or inorganic pigments; thus, the transmission spectra of the filters are determined by the absorption spectra of the materials. Using this approach, the spectral tunability is limited, and there is no way to realize well-defined bandpass or notch filters (see spectra in Table 6). For interference filters, on the other hand, spectrally tailored reflectivities up to more than 99% can be realized by tailoring the interference in the multilayer stack. This requires a high repeatability and thickness control with a tiny fraction of a wavelength, i.e., nanometer precision is needed. Thus, the requirements for each layer quality in multilayer printing are

much higher than in printing an absorptive filter since the latter usually consists of only one thick layer. To clarify the differences between the printed interference and color filters, we show a comparison in Table 6. It includes the differences in fabrication and also the differences in general filter properties.

Table 6. Differences between interference filters and color filters¹⁻¹¹.

	Dielectric interference filters	Color filters
Physical principle	Optical interference	Absorption
* Differences in fabrication*		
Layer structure	Multi thin-layer stacks	One thick layer
Typical single-layer thickness	10-300 nm	In the micrometer to millimeter range
Layer thickness control	Elaborate (on nm scale)	Rough (on sub- μm scale)
Fabrication complexity	High	Low
* Differences in the filter's property*		
Energy loss	Nearly zero	High
Heat stability	High	Low
Light stability	High	Low
Spectral profile	High degree of freedom based on design	Limited by absorbing materials
Spectral precision	Good to <1 nm	Limited by absorbing materials, > 10 nm
Spectral bandwidth (FWHM)	Narrow to <1 nm	Limited by absorbing materials, > 100 nm
Laser application suitability	High	Low
Example of a transmission spectrum		
We include this comparison in the supplementary information to emphasize this work's novelty better.

- In the manuscript, the change is

They utilize optical interference to achieve a desired transmission or reflection in the optical band of interest, which is challenging to fabricate and principally different from color filters (Supplementary Table 1).

- In the Supplementary information, a new section is added:

Comparison between optical interference filters and color filters

The novelty of this work is the successful inkjet printing of optical interference filters (OIFs), which consist of a multilayer structure. In this section, we show an overview of the difference between the OIFs and color filters, which are based on absorptive materials such as dyes or pigments.

To better illustrate the differences between the OIFs and the color filters, some major properties are included in Supplementary Table 1 for reference.

Supplementary Table 1. Differences between OIFs and color filters¹⁻¹¹.

	Dielectric interference filters	Color filters
Physical principle	Optical interference	Absorption
* Differences in fabrication*		
Layer structure	Multi thin-layer stacks	One thick layer
Typical single-layer thickness	~10-300 nm	In the micrometer to millimeter range
Layer thickness control	Elaborate (on nm scale)	Rough (on sub- μm scale)
Fabrication complexity	High	Low
* Differences in the filter's property*		
Energy loss	Nearly zero	Severe
Heat stability	High	Medium to Low
Light stability	High	Medium to Low
Spectral profile	High degree of freedom based on design	Limited by absorbing materials
Spectral precision	Good to <1 nm	Limited by absorbing materials, > 10 nm
Spectral bandwidth (FWHM)	Narrow to <1 nm	Limited by absorbing materials, > 100 nm
Laser application suitability	High	Low
Example of a transmission spectrum	 Transmittance (%) Wavelength (nm)	 TRANSMITTANCE WAVELENGTH (nm)

Comment 3:

In methods insufficient information is given about the nanoparticles. For example, what crystallographic type of TiO₂ (SiO₂) was used?

Reply to comment 3:

We thank the reviewer for the comment. To show the information of the crystallographic type of the nanoparticles, we added more information in both the manuscript and the supplementary information.

- In the manuscript, specific phase names are added:

We therefore used nanoparticulate SiO₂ (amorphous phase) and TiO₂ (rutile phase) in this work.

- In the Methods of the manuscript, new text is added:

The SiO₂ nanoparticles are in the amorphous phase, and the TiO₂ nanoparticles are in the rutile phase (Supplementary Fig. 23).

The phase components of the nanoparticles were characterized by X-ray diffraction (XRD). It was performed by a Bruker D2Phaser system with Cu-K α radiation ($\lambda = 1.5405 \text{ \AA}$) in Bragg–Brentano configuration using a LynxEye detector.

- In the Supplementary information, a new section is added:

Crystallographic of nanoparticles

X-ray diffraction analysis (XRD) was performed to characterize the crystallographic structure of the nanoparticles in the inks. The results show that the SiO₂ nanoparticles are in the amorphous phase^{12,13}. The TiO₂ nanoparticles are in the rutile phase^{14,15}. The annealing process did not change the phase of the nanoparticles.

Supplementary Fig. 23. XRD patterns of nanoparticles.

Comment 4:

Cross-section images in Fig.2 are very low resolution, but even with that on e.g. 2(b) one can see quite a lot of defects and inhomogeneities, which raises a question of reproducibility that should be addressed.

Reply to comment 4:

We thank the reviewer for the comment. To address the reproducibility concern, we reproduced 5 inkjet-printed dichroic filters with a designed center wavelength of 480 nm. Each filter consists of 18 single layers. The results show a good reproducibility of the spectrum. Therefore, we include more information on this topic.

- In the manuscript, new text is added:

Moreover, the reproducibility of the printing process was also investigated. The results indicate that the printing of OIFs is consistent and reproducible, as evidenced by the analysis of their optical performances. (Supplementary Fig. 10).

- In the Supplementary information, a new section is added:

Reproducibility of the printing process and the printed filters

To show the reproducibility of the printed filters and the printing process five dielectric filters with the same reflecting center wavelength were printed. The size of the filters is $25 \times 25 \text{ mm}^2$, and the reflection spectra were measured in the center of the filters. The light spot used in the optical measurement is 5 mm in diameter. The measured spectra are shown in Supplementary Fig. 10, where the spectra in (a) are plotted separately and (b) are plotted in one.

It can be seen from the results that all five printed optical filters show only minor differences in the optical spectrum. The deviations of the wavelengths at the reflectance of 50% are $\pm 0.5 \text{ nm}$ for the rising edge and $\pm 1 \text{ nm}$ for the falling edge. These quantitative analysis indicate that the printing results are highly reproducible.

a

b

Supplementary Fig. 10. Measured reflection spectra of five printed filters. The wavelength deviations are measured at a reflection (R) of 50%. **a** Plotted separately. **b** Plotted in one.

Comment 5:

Insufficient information is given about the produced multilayers. E.g. numbers for thicknesses of the presented layers in the films should be given.

Reply to comment 5:

We thank the reviewer for the comment. To include the necessary thickness values, we added more details in the supplementary information:

- In the Supplementary information, a new section is added:

Thickness of the designed filters

In this section, the thickness values of designed optical filters are shown.

Supplementary Table 5. Thickness of the individual layers of the longpass, shortpass, and bandpass filters.

Layer No.	Longpass filter		Shortpass filter		Bandpass filter	
	Material	Thickness (nm)	Material	Thickness (nm)	Material	Thickness (nm)
1	SiO ₂	153.20	TiO ₂	78.71	SiO ₂	153.20
2	TiO ₂	69.57	SiO ₂	119.05	TiO ₂	69.57
3	SiO ₂	70.12	TiO ₂	76.93	SiO ₂	70.12
4	TiO ₂	52.02	SiO ₂	116.96	TiO ₂	52.02
5	SiO ₂	114.15	TiO ₂	75.94	SiO ₂	114.15
6	TiO ₂	47.35	SiO ₂	117.23	TiO ₂	47.35
7	SiO ₂	87.63	TiO ₂	78.62	SiO ₂	87.63
8	TiO ₂	61.70	SiO ₂	110.40	TiO ₂	61.70
9	SiO ₂	87.63	TiO ₂	78.62	SiO ₂	87.63
10	TiO ₂	61.70	SiO ₂	110.40	TiO ₂	61.70
11	SiO ₂	87.63	TiO ₂	78.62	SiO ₂	87.63
12	TiO ₂	61.70	SiO ₂	114.09	TiO ₂	61.70
13	SiO ₂	70.13	TiO ₂	80.89	SiO ₂	70.13
14	TiO ₂	72.29	SiO ₂	110.02	TiO ₂	72.29
15	SiO ₂	92.04	TiO ₂	81.61	SiO ₂	92.04
16	TiO ₂	45.06	SiO ₂	112.45	TiO ₂	45.06
17	SiO ₂	82.13	TiO ₂	83.49	SiO ₂	82.13
18	TiO ₂	76.34	SiO ₂	120.24	TiO ₂	76.34
19			TiO ₂	83.48	SiO ₂	49.31
20					SiO ₂	118.06
21					TiO ₂	78.71
22					SiO ₂	119.05
23					TiO ₂	76.93
24					SiO ₂	116.96

25					TiO ₂	75.94
26					SiO ₂	117.23
27					TiO ₂	78.62
28					SiO ₂	110.40
29					TiO ₂	78.62
30					SiO ₂	110.40
31					TiO ₂	78.62
32					SiO ₂	78.40
33					TiO ₂	114.09
34					SiO ₂	80.89
35					TiO ₂	110.02
36					SiO ₂	81.61
37					TiO ₂	83.45
38					SiO ₂	120.24
39					TiO ₂	83.48

Supplementary Table 6. Thickness of the individual layers of the bandpass filter based on a Fabry-Pérot etalon without a silver layer.

Layer number	Material	Thickness (nm)
1	TiO ₂	77.39
2	SiO ₂	105.50
3	TiO ₂	77.39
4	SiO ₂	105.50
5	TiO ₂	77.39
6	SiO ₂	105.50
7	TiO ₂	77.39
8	SiO ₂	105.50
9	SiO ₂	105.50
10	TiO ₂	77.39
11	SiO ₂	105.50
12	TiO ₂	77.39
13	SiO ₂	105.50
14	TiO ₂	77.39
15	SiO ₂	105.50
16	TiO ₂	77.39

Supplementary Table 7. Thickness of the individual layers of the bandpass filter based on a Fabry-Pérot etalon with a silver layer.

Layer number	Material	Thickness (nm)
1	TiO ₂	77.39
2	SiO ₂	105.50
3	TiO ₂	77.39
4	SiO ₂	105.50
5	TiO ₂	77.39
6	SiO ₂	105.50
7	TiO ₂	77.39
8	SiO ₂	105.50
9	SiO ₂	70.00
10	Ag	65.00
11	SiO ₂	70.00
12	SiO ₂	105.50
13	TiO ₂	77.39
14	SiO ₂	105.50
15	TiO ₂	77.39
16	SiO ₂	105.50
17	TiO ₂	77.39
18	SiO ₂	105.50
19	TiO ₂	77.39

Supplementary Table 8. Thickness of the individual layers of dichroic filters.

		CWL380	CWL480	CWL590	CWL620	CWL680
Layer No.	Material	Thickness (nm)	Thickness (nm)	Thickness (nm)	Thickness (nm)	Thickness (nm)
1	TiO ₂	48.59	65.59	83.21	86.98	94.75
2	SiO ₂	69.20	86.20	107.15	111.89	121.72
3	TiO ₂	48.59	65.59	83.21	86.98	94.75
4	SiO ₂	69.20	86.20	107.15	111.89	121.72
5	TiO ₂	48.59	65.59	83.21	86.98	94.75
6	SiO ₂	69.20	86.20	107.15	111.89	121.72
7	TiO ₂	48.59	65.59	83.21	86.98	94.75
8	SiO ₂	69.20	86.20	107.15	111.89	121.72
9	TiO ₂	48.59	65.59	83.21	86.98	94.75
10	SiO ₂	69.20	86.20	107.15	111.89	121.72

11	TiO ₂	48.59	65.59	83.21	86.98	94.75
12	SiO ₂	69.20	86.20	107.15	111.89	121.72
13	TiO ₂	48.59	65.59	83.21	86.98	94.75
14	SiO ₂	69.20	86.20	107.15	111.89	121.72
15	TiO ₂	48.59	65.59	83.21	86.98	94.75
16	SiO ₂	69.20	86.20	107.15	111.89	121.72
17	TiO ₂	48.59	65.59	83.21	86.98	94.75
18	SiO ₂	69.20	86.20	107.15	111.89	121.72

Comment 6:

In line 196 “solar penal” should probably be corrected.

Reply to comment 6:

We thank the reviewer for the comment. The word is corrected in the manuscript.

- In the manuscript, the change is:

As an emerging use case, a building-integrated photovoltaic model based on a colored solar panel is also demonstrated.

Comment 7:

In supplementary information, in fig.10, the printed layer shows substantial inhomogeneities. There’s a significant edge defect, as well as what looks like a thickness gradient. However, authors characterize homogeneity of only a fraction of the film, which appears more homogeneous. This doesn’t appear statistically relevant.

Reply to comment 7:

We thank the reviewer for the comment. In the supplementary section entitled “Optical uniformity characterization in a 2.5×2.5 cm² size filter” the size of the characterized area is 8×8 mm². Filters with a half-inch diameter (12.5 mm) are very common on the market. These half-inch filters usually have a clear aperture of a diameter from 8 to 10 mm. Here, we give some examples of some commercial products.

Table 7. Commercial filters with a half-inch size (and smaller).

Supplier	Filter name	Stock No.	Diameter/size	Size of clear aperture
Edmund Optics	525 nm CWL bandpass	#86-939	Ø12.5 mm	8.5 mm
	450 nm Longpass	15234	6×6 mm ²	n.a.
Thorlabs	635 nm CWL bandpass	FLH05635-10	Ø12.5 mm	10 mm

In this work, we show the pristine result of the printed $2.5 \times 2.5 \text{ cm}^2$ filters without cutting the substrate. However, in practice, the printed filter needs to be cut into a size of a half-inch for a standard mounting. A homogenous area of $8 \times 8 \text{ mm}^2$ meets the commercial size requirement. Therefore, we believe the characterization is valid, however, for a half-inch-size filter. To avoid any misunderstanding, we added the essential information in the manuscript.

- In the manuscript, the change is:

Moreover, the peak, cut-on, and cut-off wavelengths were measured on nine sites of the sample to characterize the optical homogeneity in the printed bandpass filter **for a half-inch-size surface region**.

The inks designed in this work are intended for upscaling the printed filters. This can further be seen in the filters with larger sizes. For instance, $10 \times 10 \text{ cm}^2$ and A4 size filters. The edge defects did not grow with the up-scaled size, and the homogeneity was preserved, which can be seen below:

Table 8. Printed in larger size.

Printed filter		Size	$10 \times 10 \text{ cm}^2$	A4 size ($21.0 \times 29.7 \text{ cm}^2$)

Reviewer #2:

In the manuscript titled “Inkjet-Printed Optical Interference Filters”, the authors employed inkjet printing to fabricate optical interference filters, including longpass, shortpass, and bandpass filters. Compared to electron beam evaporation, plasma-assisted sputtering, and atomic layer deposition, this approach is versatile, fast, and cost-affordable. Moreover, the thin film on a large-area substrate, e.g., A4 glass substrate, showed good optical homogeneity. I recommend acceptance of the manuscript if the following issues could be addressed.

Comment 1:

What about the mechanical performance of the filter? Regarding various applications, the optical film must fulfill certain minimum mechanical requirements; these include good adhesion (evaluated using an adhesive tape peel test, frequently applied after sample exposure to a humid environment at elevated temperature), acceptable scratch and abrasion resistance to allow handling (for example, by performing a cheese cloth rubbing test).

Reply to comment 1:

We thank the reviewer for the comment. To characterize the durability of the printed OIFs, different tests based on the standards of MIL-PRF-13830B and DIN EN ISO 2409 were performed. We have included more details in the Supplementary information.

- In the manuscript, new text is added:

Moreover, our printed OIFs show good durability based on the results of adhesion and abrasion tests (Supplementary Table 3, Supplementary Table 4, and Supplementary Fig. 22).

- In Supplementary information, a new section is added:

Durability test of printed layers

To characterize the mechanical performance of the printed filter. Adhesion and moderate abrasion tests were carried out based on the standards MIL-PRF-13830B and DIN EN ISO 2409.

- Adhesion characterization (MIL-PRF-13830B and DIN EN ISO 2409)

The testing samples were two filters consisting of 4 printed layers, which were annealed at 200°C and 500°C for 10 mins, respectively. The temperature-increasing slope was 200°C per hour, and the samples were naturally cooled down to room temperature. Before performing the durability tests, the samples were placed in a test chamber with a temperature of $48 \pm 3^\circ\text{C}$ and 95% to 100% relative humidity for 24 hours (according to MIL-PRF-13830B, C.4.5.8).

Each surface was crosscut by a steel cutter. Cellophane tape was firmly pressed against the surface and quickly removed at an angle that was normal to the surface. The results are presented in the Supplementary Table 3.

Supplementary Table 3. Adhesion test based on different annealing temperatures.

Annealing	Crosscut, before applying the tape	Crosscut, after applying the tape	Area removed	Classification
200°C, 10 mins			>65%	ISO Class 5
500°C, 10 mins			5-15%	ISO Class 2

According to the results, the adhesion performance can be improved by increasing the annealing temperature. Due to the limit of the used glass substrate, the annealing temperature was limited to 500°C, and the filter adhesion classification is up to ISO Class 2.

Further increasing the adhesion between the nanoparticles and substrates, as well as the adhesion between the nanoparticles themselves, can be achieved by oligomers²⁶, including an interlayer²⁷, pre-treatment of substrate²⁸, modifying surface ligands of nanoparticles²⁹, and substrate surface roughness control³⁰.

- Moderate abrasion test (MIL-PRF-13830B, C.4.5.11)

Printed filters consisting of 4 layers were annealed at 200°C for 24 hours. Followed by being placed in a test chamber with a temperature of $48 \pm 3^\circ\text{C}$ and 95% to 100% relative humidity for 24 hours (according to MIL-PRF-13830B, C.4.5.8).

After this, the filters were rubbed with a pad of clean, dry, laundered cheesecloth (6.4 mm × 9.5 mm). The cheesecloth pad was rubbed across the surface from one point to another over the same path for 50 strokes with a force of 450 g continuously applied. The results are presented in the following Supplementary Fig. 22.

Supplementary Fig. 22. Optical images of printed filters before and after the abrasion test.

After the abrasion test, the filter did not show visible scratches according to the inspection method in Standard MIL-PRF-13830B. The induced defects can only be seen using a light microscope. The width of the scratch width is within 5 μm.

According to the standard MIL-PRF-13830B, the scratch number is used to classify the surface quality grade. The scratch number and the corresponding scratch width are in the following table.

Supplementary Table 4. Scratch number and the corresponding scratch width.

Scratch number	Scratch width (mm)
5	0.005
10	0.010
20	0.020
40	0.040
60	0.060
80	0.080
120	0.120

Based on these results, the surface quality of the printed layers is classified as Scratch number 5, denoted as the highest surface quality.

Comment 2:

The authors should compare the transmittance of the blocking range or wavelength pass range of their products with that of commercial OIFs or OIFs fabricated via vacuum-based approaches (fig. 3h). As far as I know, the optical performances of printed OIFs are still not good enough.

Reply to comment 2:

We thank the reviewer for the comment. The transmittance of the blocking or the pass region in OIFs is important. To address this point and to show a better illustration, we have included more information in the Supplementary information.

- In the manuscript, new text is added:

and enhancing the optical density by increasing the total number of layers in OIFs (Supplementary Fig. 19-21).

- In Supplementary information, a new section is added:

Comparison with commercial products

Here, we compare our inkjet-printed OIFs with the commercial products purchased from Edmund Optics. The filters compared are longpass (LP) and shortpass (SP) filters similar to our filters. However, the cut-on/off wavelengths of each compared set are not exactly the same.

The reference filters are purchased from Edmund Optics. The detailed information is listed in the following table.

Principle	Fabricating method	Filter type	Cut on/off wavelength	Size	Online store item number
Interference	Vacuum approach	LP	450 nm	6 × 6 mm ²	15234
		SP	500 nm	6 × 6 mm ²	15200

- Comparison of transmission between commercial and inkjet-printed OIFs

Compared to the commercial LP filter, which is fabricated in vacuum, the inkjet-printed LP filter shows a similar transmission in the pass region above 500 nm. The blocking behavior in the region 430-500 nm is good (approx. 3%) but lower than the commercial ones due to the lower number of layers, which can be improved by further adding layers. Furthermore, the residual transmission in the shorter wavelength range (400-450 nm) can be suppressed according to individual applications as discussed below (see subsection “Increase the optical density in inkjet-printed OIFs”).

Supplementary Fig. 19. Comparison of longpass filters.

Compared to the commercial SP filter, which is fabricated in vacuum, the inkjet-printed SP filter shows an acceptable transmission in the pass region. The transmission in the pass region can be further smoothed by optimizing layer thicknesses. Both filters have a residual transmission in the longer wavelength range, which can be further suppressed according to individual applications by more complex layer stacks.

Supplementary Fig. 20. Comparison of shortpass filters.

- Increase the optical density in inkjet-printed OIFs

It can be seen that the printed OIFs can be further optimized to achieve a higher optical density (OD) in the blocking range. To realize this, more layers can be added in future work. To better illustrate how to further improve the printed OIFs, we address the optical OD improvement as an example in this section. OD is calculated as

$$OD = -\log_{10} T$$

where T is the transmittance of the filter. OD is used to describe how well the optical power can be attenuated or blocked. A higher OD means a better capability of rejecting the light in the spectral region of low transmission. For instance, Supplementary Fig. 21 shows an LP filter requiring high transmission above 450 nm and suppressing the light transmission below 450 nm. With an increased total layer number, the OD goes up to 9. Nowadays, a typical OD for a commercial optical interference filter is between 2 and 6.

Supplementary Fig. 21. OD increases with layer number.

Comment 3:

Detailed experimental results for the thickness control of the SiO₂ or TiO₂ layers should be provided.

Reply to comment 3:

We thank the reviewer for the comment. We have elaborated on this topic in our response to comment 1 of reviewer 1. We are repeating the here the paragraphs from above.

To show the detailed method for layer thickness control, we included more experiments and discussions on the control of the layer thickness.

- In the manuscript, new text is added:

The results show a good linear dependency of the printing resolution and the deposited layer thickness, which is quantitatively indicated by a coefficient of determination of 0.9999.

- In Supplementary information, a new section is added:

Thickness control of inkjet-printed layer

The strategy of thickness control in this work is realized by controlling the total amount of ink volume deposited on a unit area. The total deposited ink volume is governed by the number of printed droplets, controlled by the printing resolution, i.e., dots per inch (dpi).

Assume the number of printed droplets on a unit area is N_0 , resulting in a thickness of T_0 . For printing a thicker layer with a thickness of T_1 , a droplet number of N_1 is needed.

$$\begin{aligned} T_1 - T_0 &= \Delta T \\ N_1 - N_0 &= \Delta N \end{aligned}$$

A well-coupled relation between ΔT and ΔN is essential to realize good thickness control.

In this work, using formulated SiO_2 ink, different ink droplet numbers N_i (400, 500, 600, 700, 800, 900 thousand) were used to print on individual substrates with the same printing area of $25 \times 25 \text{ mm}^2$. Layer thickness was measured by a profilometer (Dektak, Bruker). For each N_i , printing was performed three times to show the repeatability of the thickness, which gives a standard deviation of the thickness. The results are shown in Supplementary Fig. 9.

It can be found that, with a ΔN of 100 thousand, the thickness increases linearly with the N . The coefficient of determination is 0.9999, indicating an excellent regression. Therefore, a well-controlled thickness by changing the ΔN can be successfully achieved. Furthermore, for each printed thickness, individual deviation stays within 1% of the layer thickness, implying a highly reproducible thickness and a high repeatability of the printing process and results.

Supplementary Fig. 9. Thickness over the total amount of the printed droplet number N_i .

Comment 4:

How did the distance between the glass substrates and the printer heads affect the filter quality?

Reply to comment 4:

We thank the reviewer for the comment. The distance d between the nozzles and the substrate is crucial for achieving a homogenous printed layer.

The final thin film quality is determined not only by the formulated inks but also by factors like printing speed, drying conditions, and distance between nozzles and the substrates. We conducted the thin film printing with different d using the TiO_2 inks. The light microscope images are shown in Table 9. It can be seen that the thin film quality becomes worse with too large d values.

Table 9. Thin film quality v.s. distance between the nozzles and the substrate.

This deteriorated homogeneity can be potentially attributed to the affected ink trajectory by the aerodynamic effects caused by the airflow between the relative movement between the printing head and the substrate¹⁶. We show a figure from the referenced paper to help the illustration.

($d = 5 \text{ mm}$, adopted from the reference¹⁶)

In our experiments, the distance d was 1.3 mm, which was empirically optimized at the beginning of the study.

Comment 5:

The transmitting spectra of bare glass substrate should also be provided in the manuscript.

Reply to comment 5:

We thank the reviewer for the comment. To show the transmittance and the reflectance of the glass substrate, measurement results are added in the supplementary information.

- In Supplementary information, a new section is added:

Transmittance of glass substrate

Supplementary Fig. 24. Transmittance and reflectance curve of glass substrate.

Comment 6:

The authors mentioned that inkjet printing can be applied on large area and flexible substrates under ambient conditions. Since the annealing processes were performed using a hotplate with the temperature at 200 °C for 10 min, I wonder if the proposed method can be employed to fabricate flexible filters.

Reply to comment 6:

We thank the reviewer for the comment. We performed the developed printing process on a PET foil. The difference between printing on a foil and a glass substrate is the post-treatment, i.e., the annealing temperature and annealing time. We used 100°C and 1 hour instead of 200°C and 10 min for the post-annealing in order to avoid damage to the foil. The image below shows the patterned filter printed on a flexible PET foil. As a proof of concept, the total layer number of the printed OIFs is 8.

- In the manuscript, new text is added:

Additionally, we exhibited the versatility of this developed method by fabricating a flexible and patterned filter on a foil, achieved by straightforward adjustments of the process parameters (Supplementary Fig. 17).

- In Supplementary information, a new section is added:

Printing OIFs on flexible foil

To show the possibility of applying the developed printing process on a flexible substrate. Patterned OIFs were designed and printed on PET foil (Puetz Folien). The size of the foil substrate is $9 \times 9 \text{ cm}^2$.

The parameters in the printing remained the same. The changed parameters are the annealing temperature and the time. The annealing temperature was reduced from 200°C to 100°C, and the annealing time increased from 10 min to 1 hour for each printed single layer. As a proof of concept, the total layer number of the printed OIFs is 8 in Supplementary Fig. 17.

Supplementary Fig. 17. Patterned OIFs printed on a flexible PET foil.

Comment 7:

The authors should clarify if the optical multilayer thin films can be improved to eliminate the miscellaneous transmittance peaks of the OIFs, for example, longpass filter (400-450 nm) , shortpass filter (600-800 nm).

Reply to comment 7:

We thank the reviewer for the comment. To adequately address this, we include more information in the supplementary information.

- In the manuscript, new text is added:

Subsequently, the strategy to further improve the quality of the inkjet-printed filters is given. For instance, the suppression of residual transmittance in edge and bandpass filters via the integration of additional optical cavities (Supplementary Fig. 18)

- In Supplementary information, a new section is added

Residual transmission elimination

In optical filter design, expanding the optical stopband is a common strategy to eliminate miscellaneous transmission. This can be realized by stacking multiple optical cavities. In this work, the printed longpass filter has a cut-on wavelength of 532 nm. To suppress the miscellaneous transmittance in the 400 to 450 nm wavelength range, a second cavity behaving as another longpass filter can be stacked on the original cavity.

Here, we show a proof of concept of suppressing the undesired transmission. Supplementary Fig. 18a shows the designed single cavity longpass filter in this work, namely the original cavity. Supplementary Fig. 18b shows the transmission curve of a second cavity. By stacking these two cavities, the residual transmission between 400 and 450 nm can be suppressed, as shown in Supplementary Fig. 18c. Furthermore, optimization of individual layer thickness and the total layer number of each optical cavity can be applied based on the new stack.

The strategy shown above can be used to eliminate undesired transmission in longpass, shortpass, and bandpass filters.

a

b

c

Supplementary Fig. 18. Transmittance of filters with different designs.

References

1. Asghar, M. H., Shoaib, M., Placido, F. & Naseem, S. Modeling and preparation of practical optical filters. *Current Applied Physics* **9**, 1046–1053; 10.1016/j.cap.2008.11.007 (2009).
2. Martinu, L. & Poitras, D. Plasma deposition of optical films and coatings: A review. *Journal of Vacuum Science & Technology A: Vacuum, Surfaces, and Films* **18**, 2619–2645; 10.1116/1.1314395 (2000).
3. *Photonic Crystal and Its Applications for Next Generation Systems*. 1st ed. (Springer Nature Singapore, Singapore, 2023).
4. Musgraves, J. D., Hu, J. & Calvez, L. *Springer handbook of glass* (Springer, Cham, Switzerland, 2019).
5. *LICHT 2016. Karlsruhe, 25. - 28. September ; Tagungsband - Proceedings ; [22. Gemeinschaftstagung = 22nd Associations' Meeting]* (KIT Scientific Publishing, Karlsruhe, 2016).
6. Stone, M. C. *A field guide to digital color* (A K Peters/CRC Press, Place of publication not identified, 2016).
7. Baumeister, P. *Optical coating technology* (SPIE Optical Engineering Press, Bellingham WA., 2004).
8. Karim, M. A. *Electro-optical displays*. 1st ed. (CRC Press, Boca Raton, 2020).
9. Zapka, W. & Zapka, W. e. *Handbook of industrial inkjet printing. A full system approach / Werner Zapka*. 1st ed. (Wiley-VCH, Weinheim, 2017).
10. Träger, F. *Springer handbook of lasers and optics*. 2nd ed. (Springer, Dordrecht, New York, 2012).
11. Macleod, H. A. *Thin-film optical filters*. 4th ed. (CRC; London : Taylor & Francis [distributor], Boca Raton, Fla., 2010).
12. Nayak, P. P. & Datta, A. K. Synthesis of SiO₂-Nanoparticles from Rice Husk Ash and its Comparison with Commercial Amorphous Silica through Material Characterization. *Silicon* **13**, 1209–1214; 10.1007/s12633-020-00509-y (2021).
13. Sompech, S., Dasri, T. & Thaomola, S. Preparation and Characterization of Amorphous Silica and Calcium Oxide from Agricultural Wastes. *Orient. J. Chem* **32**, 1923–1928; 10.13005/ojc/320418 (2016).
14. Mayabadi, A. H. *et al.* Evolution of structural and optical properties of rutile TiO₂ thin films synthesized at room temperature by chemical bath deposition method. *Journal of Physics and Chemistry of Solids* **75**, 182–187; 10.1016/j.jpccs.2013.09.008 (2014).
15. You, Y. F. *et al.* Structural characterization and optical property of TiO₂ powders prepared by the sol–gel method. *Ceramics International* **40**, 8659–8666; 10.1016/j.ceramint.2014.01.083 (2014).
16. Rodriguez-Rivero, C., Castrejón-Pita, J. R. & Hutchings, I. M. Aerodynamic Effects in Industrial Inkjet Printing. *jist* **59**, 40401-1-40401-10; 10.2352/J.ImagingSci.Technol.2015.59.4.040401 (2015).

REVIEWER COMMENTS

Reviewer #1 (Remarks to the Author):

The revised paper contains substantial amount of additional information, which unfortunately makes it very hard to navigate and find specific information.

It is certainly appreciated that it takes substantial effort to optimize printing parameters, as well as ink formulations for a specific application, especially, as in this case, when control of thickness for many layers in a multilayer sandwich is required. However, part of the optimization parameters are printhead related, so for a different printer further optimization will be required.

In their rebuttal letter the authors claim that they have nm control of thickness for every layer and then list thickness values in nm with two digits after the decimal point and sub-nanometer error bars. At the same time they enclose an AFM image where they show RMS roughness of 5.6nm and peak-to-valley of ~50 nm, so locally there's more than several nm deviations of thickness. Further, they show a thickness calibration line with sub-nm error bars (supplementary fig.9) and a set of tables with thickness values from optical measurements (section 20 of the supplementary data), which deviate from the calibration line by more than the error bars. In section 9 of the same file the authors show the reproducibility of the transmittance data between 5 samples, so it appears that their filters are not very sensitive to some thickness variations of the layers (although it's unclear to which extent of such variation).

I have trouble understanding the description in section 8 "Thickness control of inkjet-printed layer". Specifically, what is implied by "droplet numbers". If this is the total amount of droplets in the printed layer, then for e.g. for $N=400$ and printed area of $2.5 \times 2.5 \text{ cm}^2$ this would imply printing density of 0.64 drop/mm^2 , which cannot result in 57.5 nm thickness at 10pL per droplet, especially at 3% concentration, from simple volume considerations. This needs to be clarified.

I did not find a mention of the light incident angle used for measurements, which would affect the peak wavelength. I'm assuming it's normal to the surface.

The authors replied to my comment about inhomogeneity of their $2.5 \times 2.5 \text{ cm}^2$ size filter shown in Fig 12 of the revised supplementary data: "Filters with a half-inch diameter (12.5 mm) are very common on the market. These half-inch filters usually have a clear aperture of a diameter from 8 to 10 mm". This is certainly true. However, I strongly suspect that 12.5 mm diameter devices made by inkjet would be even less homogeneous than the $2.5 \times 2.5 \text{ cm}^2$ ones.

The name of the section 18 of the supplementary data should probably be corrected.

Reviewer #2 (Remarks to the Author):

The authors have answered all my concerns. Thus I recommend the publication in Nature Communications.

Dear reviewer,

In this letter, we respond to the comments point by point.

- The original comments are in black
- The replies to the comments are in blue
- The changes made in the manuscript and supplementary information are highlighted.

Reviewer #1:

Comment 1:

The revised paper contains substantial amount of additional information, which unfortunately makes it very hard to navigate and find specific information.

Reply to comment 1:

We thank the reviewer for the comment. Hereby, we have adjusted the order of some sections and added a table at the beginning of the supplementary information to facilitate navigation and help locate specific information. The added table to the supplementary information is:

The following table summarizes the key information of the sections

Section 1:	Highlighting the difference between "multilayer interference filter" and "color filters"
Section 2-4:	Ink development related
Section 5:	Refractive index of the materials
Section 6-9:	Printing process optimization and layer quality related
Section 10-15:	Properties of inkjet-printed optical interference filters
Section 16-17:	Performance enhancement and comparison with commercial products
Section 18:	Durability test according to standards
Section 19:	Crystal structures of nanoparticles
Section 20:	Designed thickness values of layers
Section 21:	Transmittance curve of the glass substrate

Comment 2:

It is certainly appreciated that it takes substantial effort to optimize printing parameters, as well as ink formulations for a specific application, especially, as in this case, when control of thickness for many layers in a multilayer sandwich is required. However, part of the optimization parameters are printhead related, so for a different printer further optimization will be required.

Reply to comment 2:

We appreciate that the reviewer pointed out this because it is important. Indeed, using different devices and printheads would require different printing parameters. Therefore, further adjustments on these parameters need to be conducted when transferring the developed printing process to different printheads or, even more importantly, to an industrial printer. In this work, we demonstrated the proof of concept that optical interference filters are inkjet-printable. The widely-used printhead type for

functional materials, the FUJIFILM DMC cartridges, was used for demonstration. This research-type printhead has an upscale industrial printhead version, the Spectra Q-Class printhead, where the developed process and ink formulations could be transferred in the future. While the details of the printing parameters will be different, the general approach toward optimization can be very similar. We, therefore, believe that sharing the printing parameters and approach is helpful and stimulating for the readership.

Comment 3:

In their rebuttal letter the authors claim that they have nm control of thickness for every layer and then list thickness values in nm with two digits after the decimal point and sub-nanometer error bars. At the same time they enclose an AFM image where they show RMS roughness of 5.6 nm and peak-to-valley of ~50 nm, so locally there's more than several nm deviations of thickness.

Reply to comment 3:

We thank the reviewer for the comment. The AFM image shows an overall surface roughness for a 39-layer stack. In contrast, in *Supplementary Fig. 9*, each thickness corresponds to a respective single layer. The thickness measurement is conducted by scanning the surface over a length of 0.5 mm, and the layer thickness is an average value out of multiple local points. The displayed thickness values in *Supplementary Fig. 9* are further statistical results in the form of an averaged value out of repeated measurements. The deviations between the repeated measurement results were very small, therefore resulting in a very small error bar in the diagram. In other words, layers (repeatedly) printed with the same parameters show very similar thicknesses, indicating a repeatable printing process and consistent results. Hence, the data and our statement are not contradictory.

The details of roughness accumulation and intrinsic smoothening are the subject of further investigation. The already convincing optical properties, however, indicate that the roughness is not a major problem of our approach.

To address the concern, we have included the corresponding atomic force microscopy images in the supplementary information to share the relevant data. The newly added section in the supplementary information is:

9. Surface roughness of the printed layers

Details of the roughness of the layers were investigated by atomic force microscopy (AFM).

Supplementary Fig. 1. Surface roughness characterized by AFM. **a** Printed SiO₂ Single layer. Thickness \approx 100 nm. **b** Printed TiO₂ Single layer. Thickness \approx 100 nm. **c** Printed 39-layer stack. Thickness \approx 3.3 μm .

To clarify the method of thickness determination, in the Supplementary information **Section 8. Thickness control of inkjet-printed layer**, we added:

Each thickness was determined as the average value of multiple sites along the measuring path (0.5 mm long).

In addition, we added the following sentence to the manuscript:

Surface roughness of the printed layers was characterized by atomic force microscope, showing low root-mean-square roughness values (Supplementary Fig. 10).

Furthermore, in the manuscript METHOS, we added:

The surface morphology was measured by atomic force microscope (AFM) (NanoWizard, Bruker Nano), and the surface RMS roughness was analyzed with software Gwyddion.

Comment 4:

Further, they show a thickness calibration line with sub-nm error bars (supplementary fig.9) and a set of tables with thickness values from optical measurements (section 20 of the supplementary data), which deviate from the calibration line by more than the error bars. In section 9 of the same file the authors show the reproducibility of the transmittance data between 5 samples, so it appears that their filters are not very sensitive to some thickness variations of the layers (although it's unclear to which extent of such variation).

Reply to comment 4:

We thank the viewer for the comment. Unfortunately, there is the misunderstanding that the layers consist of same material were all supposed to have the same thicknesses in the filter. This is not the case.

- In *Supplementary Fig. 9*, the individual thickness corresponds to a printed single layer, demonstrating effective control over layer thickness.
- In contrast, *Supplementary Section 20* provides the specific calculated thicknesses for different layers in filters obtained from the simulation software. Especially in the nonperiodic structures, the layer thicknesses are designed and meant to be different for achieving desired spectral performance. Therefore, this is different information with no conflict to *Supplementary Fig. 9*.

For example, the individual layer thicknesses of the longpass filter are shown in the following table (copied from *Supplementary Section 20*). The filter was designed to have a **non-periodic** structure, i.e., each layer has its own optimized thickness, to ease the ripples (seen in the figure below, copied from *Supplementary Section 11*). For individual layers in this longpass filter, the effective thickness control presented in *Supplementary Fig. 9* would then be essential to reach the target thicknesses.

Longpass filter		
Layer No.	Material	Thickness (nm)
1	SiO ₂	153.20
2	TiO ₂	69.57
3	SiO ₂	70.12
4	TiO ₂	52.02
5	SiO ₂	114.15
6	TiO ₂	47.35
7	SiO ₂	87.63
8	TiO ₂	61.70
9	SiO ₂	87.63
10	TiO ₂	61.70
11	SiO ₂	87.63
12	TiO ₂	61.70
13	SiO ₂	70.13
14	TiO ₂	72.29
15	SiO ₂	92.04
16	TiO ₂	45.06
17	SiO ₂	82.13
18	TiO ₂	76.34

We have included additional statements in the supplementary information to clarify this issue.

In Section 11. Comparison of ripples in the transmitting curve between periodic and non-periodic structures, we have added:

Therefore, the thickness of each layer can be different from others. Details can be found in Section 20.

In Section 20. Designed thickness of the filters, we revised the following sentence:

In this section, the designed thicknesses of optical filters are shown. The values are obtained from the simulation software Essential Macleod.

Besides, we thank the reviewer for pointing out that the reproducibility of the filter is satisfactory.

Comment 5:

I have trouble understanding the description in section 8 “Thickness control of inkjet-printed layer”. Specifically, what is implied by “droplet numbers”. If this is the total amount of droplets in the printed layer, then for e.g. for $N=400$ and printed area of $2.5 \times 2.5 \text{ cm}^2$ this would imply printing density of 0.64 drop/mm^2 , which cannot result in 57.5 nm thickness at 10 pL per droplet, especially at 3% concentration, from simple volume considerations. This needs to be clarified.

Reply to comment 5:

We thank the reviewer for the comment.

The relevant content in the last supplementary information is as follows:

“In this work, using formulated SiO₂ ink, different ink droplet numbers N_i (400, 500, 600, 700, 800, 900 thousand) were used to print on individual substrates with the same printing area of 25 × 25 mm².”

Therefore, the N mentioned by the reviewer is 400 thousand instead of 400, leading to the density of 400000 droplets/area. To improve the clarity, this sentence is revised as below:

In this work, using formulated SiO₂ ink, different ink droplet numbers N_i (400k, 500k, 600k, 700k, 800k, 900k, where 400k denotes 400000 and the same rule applies for other numbers) were used to print on individual substrates with the same printing area of 25 × 25 mm².

Another relevant sentence is also revised as below:

It can be found that the thickness increases linearly with the N.

Comment 6:

I did not find a mention of the light incident angle used for measurements, which would affect the peak wavelength. I'm assuming it's normal to the surface.

Reply to comment 6:

We thank the reviewer for the comment. We agree that these are essential details, and we have added the following sentence in the METHODS section of the manuscript:

“The transmittance was obtained with the normal incident angle, the reflectance was obtained with 8° incident angle.”

Here, the 8° incident angle is due to the setup.

Comment 7:

The authors replied to my comment about inhomogeneity of their 2.5 × 2.5 cm² size filter shown in Fig 12 of the revised supplementary data: “Filters with a half-inch diameter (12.5 mm) are very common on the market. These half-inch filters usually have a clear aperture of a diameter from 8 to 10 mm”. This is certainly true. However, I strongly suspect that 12.5 mm diameter devices made by inkjet would be even less homogeneous than the 2.5 × 2.5 cm² ones.

Reply to comment 7:

We thank the reviewer for the comment. The reviewer is correct. Using a 2.5 × 2.5 cm² printing area, a 12.5 mm diameter homogenous area has been achieved. Furthermore, when using a 12.5 mm diameter area, a homogenous area with a smaller diameter can be obtained. This is limited by the edge defects. However, in this work, we aim to demonstrate the upscaling of the functional filters. Moreover, the ink formulation will need to be adapted for small-sized filters compared to the results in this work.

We agree with the reviewer that this can be a very interesting topic, and we would like to investigate in this direction in future research.

In this work, we have demonstrated our developed ink formulation and printing process with a focus on large-size filters. There is, however, the possibility to cut individualized sizes from the larger area filters. This possibility is now also mentioned in the manuscript:

We mention that this also opens the possibility to cut customized smaller size filters from the larger area, e.g., by laser cutting.

Comment 8:

The name of the section 18 of the supplementary data should probably be corrected.

Reply to comment 8:

We thank the reviewer for the comment. In the updated Supplementary information file, we have revised the title as:

19. Crystal structure characterization of nanoparticles

The change of number from 18 to 19 is due to the newly added section 9.

REVIEWERS' COMMENTS

Reviewer #1 (Remarks to the Author):

I have no further comments to this manuscript.